Fusion neural networks for plant classification: learning to combine RGB, hyperspectral, and lidar data

http://orcid.org/0000-0002-2085-1449 Scholl Victoria M. 1 2 Victoria.Scholl@colorado.edu
McGlinchy Joseph 1
Price-Broncucia Teo 3
http://orcid.org/0000-0002-3983-7970 Balch Jennifer K. 1 2
http://orcid.org/0000-0002-7745-9990 Joseph Maxwell B. 1
1 Earth Lab, Cooperative Institute for Research in Environmental Science, University of Colorado at Boulder , Boulder, Colorado , United States
2 Department of Geography, University of Colorado at Boulder , Boulder, Colorado , United States
3 Department of Computer Science, University of Colorado at Boulder , Boulder, Colorado , United States
Mortimer Monika
Electronic publication date: 2021 Jul 29
Publication date: 2021
Volume: 9
Electronic Location ID: e11790
Received 2021 Jan 27; Accepted 2021 Jun 25
Copyright: © 2021 Scholl et al.
Copyright year: 2021
Copyright holder: Scholl et al.
License: This is an open access article distributed under the terms of the Creative Commons Attribution License, which permits unrestricted use, distribution, reproduction and adaptation in any medium and for any purpose provided that it is properly attributed. For attribution, the original author(s), title, publication source (PeerJ) and either DOI or URL of the article must be cited.
License URL: https://creativecommons.org/licenses/by/4.0/

Keywords: Machine learning, Deep learning, Species classification, Remote sensing, Airborne remote sensing, National Ecological Observatory Network, Data science competition, Neural networks, Open science

Funding: Earth Lab Funding for this work was provided by Earth Lab, through the University of Colorado at Boulder (CU Boulder) Grand Challenge Initiative, and the Cooperative Institute for Research in Environmental Sciences (CIRES) at CU Boulder. The funders had no role in study design, data collection and analysis, decision to publish, or preparation of the manuscript.

==============================
Airborne remote sensing offers unprecedented opportunities to efficiently monitor vegetation, but methods to delineate and classify individual plant species using the collected data are still actively being developed and improved. The Integrating Data science with Trees and Remote Sensing (IDTReeS) plant identification competition openly invited scientists to create and compare individual tree mapping methods. Participants were tasked with training taxon identification algorithms based on two sites, to then transfer their methods to a third unseen site, using field-based plant observations in combination with airborne remote sensing image data products from the National Ecological Observatory Network (NEON). These data were captured by a high resolution digital camera sensitive to red, green, blue (RGB) light, hyperspectral imaging spectrometer spanning the visible to shortwave infrared wavelengths, and lidar systems to capture the spectral and structural properties of vegetation. As participants in the IDTReeS competition, we developed a two-stage deep learning approach to integrate NEON remote sensing data from all three sensors and classify individual plant species and genera. The first stage was a convolutional neural network that generates taxon probabilities from RGB images, and the second stage was a fusion neural network that “learns” how to combine these probabilities with hyperspectral and lidar data. Our two-stage approach leverages the ability of neural networks to flexibly and automatically extract descriptive features from complex image data with high dimensionality. Our method achieved an overall classification accuracy of 0.51 based on the training set, and 0.32 based on the test set which contained data from an unseen site with unknown taxa classes. Although transferability of classification algorithms to unseen sites with unknown species and genus classes proved to be a challenging task, developing methods with openly available NEON data that will be collected in a standardized format for 30 years allows for continual improvements and major gains for members of the computational ecology community. We outline promising directions related to data preparation and processing techniques for further investigation, and provide our code to contribute to open reproducible science efforts.

Introduction

Understanding the species composition of individual trees within forests is essential for monitoring biodiversity (Nagendra, 2001; Wang et al., 2010), invasive species (Asner et al., 2008; He et al., 2011), terrestrial carbon (Schimel et al., 2015; Jucker et al., 2017), and disturbance regimes (Kulakowski, Veblen & Bebi, 2003; Senf, Seidl & Hostert, 2017). Remote sensing enables us to more efficiently map and monitor vegetation than using traditional field-based methods alone, using platforms ranging in scale from drones to satellites carrying a wide variety of sensors (Kerr & Ostrovsky, 2003; White et al., 2016; Lucash et al., 2018). Different types of passive and active imaging sensors provide unique information about ecosystems that may be most useful when combined (Anderson et al., 2008; Tusa et al., 2020). Multispectral cameras are accessible, affordable and typically require minimal post-processing to be ready for analysis (Gini et al., 2018; Abdollahnejad & Panagiotidis, 2020). Hyperspectral data are valuable for their ability to capture spectral signatures beyond the visible wavelengths, which often contain descriptive reflectance characteristics across plant types and conditions (Dalponte et al., 2012; Ballanti et al., 2016). Active sensors such as Light Detection and Ranging (lidar) emit pulses of laser light and record the amount and intensity of reflected energy. Lidar data provide structural information about the height, shape, and variability of tree crowns (Heinzel & Koch, 2011; Koenig & Höfle, 2016).

Any one data source could be used for plant species classification, but combining information from multiple sources is valuable, albeit difficult (Torabzadeh, Morsdorf & Schaepman, 2014; Anderson et al., 2008; Asner et al., 2012). Deep neural networks automatically extract intricate patterns and identify trends from large volumes of data (LeCun, Bengio & Hinton, 2015), which makes them useful for classification and data fusion tasks (Zhu et al., 2017; Ma et al., 2019), including plant species classification (Brodrick, Davies & Asner, 2019; Fricker et al., 2019; Zhang et al., 2020; Onishi & Ise, 2021). At a high level, neural networks are flexible function approximators that learn a mapping from inputs (e.g., spectral or lidar data) to outputs (e.g., species classes), by way of a sequence of matrix multiplications and nonlinearities. By providing different kinds of input to the same network (e.g., a multimodal network that ingests spectral and lidar data), neural networks learn how to fuse different data sources, in contrast to more manual approaches in which a human analyst decides how to combine disparate data ahead of time (Diaz et al., 2020).

Here we describe the deep learning classification approach used by the Jeepers Treepers team for the Integrating Data science with Trees and Remote Sensing (IDTreeS) 2020 plant classification challenge (https://idtrees.org). IDTReeS uses publicly available data from the National Ecological Observatory Network (NEON), funded by the National Science Foundation (NSF) to measure long-term ecological change at 81 field sites in 20 ecoclimatic domains across the United States, Alaska, Hawaii and Puerto Rico (Keller et al., 2008). The NEON data provided for this competition include both field-based plant measurements and airborne remote sensing data products derived from high resolution red, green, blue (RGB) digital camera imagery, hyperspectral imagery across the visible to shortwave infrared wavelengths, and light detection and ranging (lidar) data (Johnson et al., 2010). By participating in an open competition, teams are encouraged to innovate and accelerate their computational methods development (Carpenter, 2011). An earlier iteration of this competition used NEON data from a single forest to convert images into information on individual trees (Marconi et al., 2019), while this 2020 competition used data from three sites to compare how transferable teams’ methods were to unseen sites. Classifier transferability to out-of-sample spatial, temporal, and geographic regions is particularly important in cases where data are limited (Wu et al., 2006; Moon et al., 2017). In addition to emphasizing method generalization across sites, this competition tasked teams with designing classification models that can deal with species and genera from outside of the training set. We begin with a description of our data processing steps, then segue into a three stage classification pipeline, and finally report our results along with ideas for future investigation. All of the data and processing tools that we used are open source and publicly available to support and enable reproducible science.

Methods

Study area

The IDTReeS competition included data from three different NEON domains in the southeastern United States, each with distinct ecological and climatic characteristics (Fig. 1). Ordway-Swisher Biological Station (OSBS) in Florida in the Southeast NEON domain features a mixed forest of hardwood and conifers, mostly dominated by pines. Mountain Lake Biological Station (MLBS) in Virginia in the Appalachians and Cumberland Plateau NEON domain is mainly composed of hardwood trees. Talladega National Forest (TALL) in Alabama in the Ozarks complex NEON domain is dominated by mixed hardwoods and conifers (mostly pine), with a tree species composition that is largely a mixture of species found in OSBS and MLBS. Training data for the competition were provided at two of the three NEON sites, MLBS and OSBS, and then our classification method was evaluated at those two sites in addition to the TALL site (where our classifier has not seen data).

Figure 1 Study Area. The data provided for the tree mapping competition belong to three National Ecological Observatory Network (NEON) sites: Ordway-Swisher Biological Station (OSBS) in Florida, Mountain Lake Biological Station (MLBS) in Virginia, and Talladega National Forest (TALL) in Alabama.

The sites span three separate NEON domains in the southeastern United States, each with distinct ecological and climatic characteristics, although the TALL site has a species composition that is largely a mixture of species found in OSBS and MLBS.

Data processing

The IDTReeS research team provided publicly available geospatial and tabular data for use in this competition (Graves & Marconi, 2020). We processed raw NEON data to generate a feature vector for each individual plant canopy, then passed these vectors to a multimodal neural network (Ngiam et al., 2011) that ultimately makes the taxon predictions (Fig. 2). The raw geospatial data that we used include high resolution orthorectified red, green, blue (RGB) digital camera imagery with 10 cm spatial resolution (NEON.DP1.30010), hyperspectral reflectance data from the NEON Imaging Spectrometer with 1 m resolution (NEON.DP3.30006), and discrete lidar point cloud data with a point density of ≈3.15 points per square meter (NEON.DP1.30003) provided by the IDTreeS competition at each of the NEON ground plots (Graves & Marconi, 2020; NEON, 2020).

Figure 2 The workflow. We processed raw remote sensing data products into formats that describe the spectral and structural characteristics for each individual plant canopy.

We used a pre-trained Convolutional Neural Network (ConvNet) to estimate taxon probabilities using the red, green, blue (RGB) cropped canopy images, and combined these taxon probabilities with hyperspectral reflectance spectra and lidar-derived pseudo-waveforms into a concatenated feature vector. This feature vector was the input to the so-called “fusion network”, a 2-layer multilayer perceptron (MLP) with two hidden layers (size 64 and 32) and trained using a custom “soft F1” loss function, to predict taxon class probabilities for each individual plant. We then applied post-processing including a threshold to assign individuals to an “other” class when the classification confidence was low. Finally, we produced predictions of taxon probabilities.

These data products are derived from both active and passive remote sensing systems onboard the NEON Airborne Observation Platform (AOP) to capture the structural and spectral characteristics of vegetation (Kampe et al., 2010a). The high resolution RGB images are collected by an Optech D8900 digital color camera and capture fine spatial details of the tree crowns across the visible wavelengths (Gallery et al., 2015). The hyperspectral reflectance data have 426 spectral bands spanning the visible to infrared regions from 380–2,510 nm in increments of 5 nm (Karpowicz & Kampe, 2015). The lidar data points representing the x, y, z location of surface features and the ground in three-domensional space were acquired by the Optech Incorporated Airborne Laser Terrain Mapper Gemini instrument with a near infrared laser that operates at 1,064 nm (Krause & Goulden, 2015). The NEON AOP flies at typical altitude of 1,000 m above ground level and is intended to collect airborne data at each NEON site’s peak phenological greenness. These geospatial data products were provided in 20 m × 20 m tiles representing the size of individual sampling plots.

Woody plant vegetation structure field data (NEON.DP1.10098) in tabular form collected based on NEON’s Terrestial Observation System protocol (Thorpe et al., 2016) were provided as well, contributing information on individual tree identifiers, sampling locations, and taxonomic species or genera labels. Individual tree crown delineations were generated and provided by the IDTReeS competition research group for the classification task. Each canopy polygon was a rectangular bounding box that represents the maximum crown extent for each individual tree. Each canopy polygon was associated with a record in the NEON field data. We extracted data independently for each mapped tree canopy. First we generated a rectangular RGB image subset for each individual plant by using the provided canopy polygons to crop the RGB image tiles. Then we extracted hyperspectral reflectance data from the spatial centroid pixel within each canopy polygon. Finally, we generated pseudo-waveforms from lidar data by computing the density of point cloud returns within the boundary of each canopy polygon using 39 vertical bins of height. We split the provided data randomly into a training (75%) and initial validation (25%) set so that each individual tree was associated with just one of the data partitions. Note that we used this initial validation set to help tune our RGB classification step in the first stage of our approach. For the final evaluation of our classification method, we were provided with an independent set of data without taxa labels.

Feature extraction from RGB data

We used the cropped rectangular RGB canopy images as input to fine-tune a convolutional neural network (CNN) pretrained on the ImageNet dataset (Deng et al., 2009). CNNs have been shown effective in classification of high resolution remote sensing images by learning textural and spatial relationships through many stages of convolutional filters and pooling layers (Zhu et al., 2017).

We split the individual tree RGB canopy images randomly into training (80%) and validation (20%) subsets to tune the CNN. The RGB data consisted of 1,052 individuals each belonging to one of 31 taxa (Table 1). There was notable class imbalance; approximately one-third of the trees were PIPA2 (Pinus palustris, longleaf pine) while many species or genera only had one or two samples represented in the data set.

Table 1 Taxa included in the training data.

Each row represents a unique class for the classifier.

Taxon code	Scientific name	Common name	Count	
PIPA2	Pinus palustris	longleaf pine	237	
QURU	Quercus rubra	northern red oak	138	
ACRU	Acer pensylvanicum	striped maple	104	
QUAL	Quercus alba	white oak	86	
QULA2	Quercus laevis	turkey oak	59	
QUCO2	Quercus coccinea	scarlet oak	39	
AMLA	Amelanchier laevis	Allegheny serviceberry	38	
NYSY	Nyssa sylvatica	blackgum	33	
LITU	Liriodendron tulipifera	tuliptree	16	
QUGE2	Quercus geminata	sand live oak	15	
MAGNO	Magnolia sp.	magnolia	12	
QUMO4	Quercus montana	chestnut oak	10	
OXYDE	Oxydendrum sp.	sourwood	9	
BETUL	Betula sp.	birch	6	
PINUS	Pinus sp.	pine	6	
PRSE2	Prunus serotina	black cherry	6	
ACPE	Acer rubrum	red maple	5	
PIEL	Pinus elliottii	slash pine	4	
CAGL8	Carya glabra	pignut hickory	3	
FAGR	Fagus grandifolia	American beech	3	
PITA	Pinus taeda	loblolly pine	3	
QUHE2	Quercus hemisphaerica	Darlington oak	3	
ROPS	Robinia pseudoacacia	black locust	2	
TSCA	Tsuga canadensis	eastern hemlock	2	
ACSA3	Acer saccharum	sugar maple	1	
CATO6	Carya tomentosa	mockernut hickory	1	
GOLA	Gordonia lasianthus	loblolly bay	1	
LYLU3	Lyonia lucida	fetterbush lyonia	1	
NYBI	Nyssa biflora	swamp tupelo	1	
QUERC	Quercus sp.	oak	1	
QULA3	Quercus laurifolia	laurel oak	1	

The size and dimensions of the rectangular canopy polygons were quite variable (Fig. 3). Since the pretrained CNN requires each image to have the same dimensions, we transformed each rectangular RGB canopy cropped image to be 224 × 224 pixels using a combination of cropping and resizing, and each image was normalized based on the mean and standard deviation of the ImageNet data set. We labeled each of the resized and normalized RGB canopy images with its respective taxon identification code (Fig. 4).

Figure 3 We used the individual tree crown rectangular polygons to clip remote sensing image layers, such as the 10 cm high spatial resolution red, green, blue (RGB) data shown here at the (A) Ordway-Swisher Biological Station (OSBS) and (B) Mountain Lake Biological Station (MLBS) sites.

Figure 4 Nine corresponding pairs of RGB image chips, cropped using individual tree crown polygons, with their original crown dimensions (A) and after being resized to 224 × 224 pixels (B) to yield consistently shaped inputs for the ResNet classifier. Each image chip is labeled with the taxon identification code that corresponds to each individual plant’s scientific name.

ResNet evaluation

We tested a series of ResNet CNNs (He et al., 2016) to generate a probability for each taxon class from the RGB image chips. We loaded pretrained weights generated from the ImageNet dataset, using ResNets that varied in depth including architectures with 18, 34, 50, 101, and 152 layer encoders. We compared these different depth ResNets in terms of the macro F1 score, precision, and recall using our validation subset of the RGB data. The summary of these values is presented in Table 2.

Table 2 Macro F1 score, precision, and recall values for different ResNet convolutional neural network (CNN) architectures that we tested for the red, green, blue (RGB) image classifier.

Encoder layers	F1	Precision	Recall	
18	0.4282	0.3408	0.1642	
34	0.4698	0.1909	0.1392	
50	0.3463	0.2098	0.1228	
101	0.465	0.2916	0.1528	
152	0.3867	0.2635	0.1571	

In a 2-class problem, precision is the proportion of positive predictions which are actually correct, whereas recall is identifies the proportion of actual positive predictions which are correct. F1 score is the harmonic mean of both precision and recall and was an evaluation metric in the competition. To compute the multi-class value of precision, recall, and F1-score, we computed the average across all classes. The Resnet-34 had the highest F1 score, and was used to generate RGB features for the fusion model.

Pseudo-waveform generation from lidar point cloud

The lidar point cloud contains information on the 3-dimensional structure of tree canopies (Dubayah & Drake, 2000; Kampe et al., 2010b). As the laser travels through the canopy during lidar data collection, energy within the beam’s footprint is reflected by the top of the canopy, interactions with sub-canopy elements, and potentially the underlying terrain surface. The returned energy waveform is sampled to produce multiple “returns’’, points describing the spatial and vertical vegetation structure (Lefsky et al., 2002). The precise 3-dimensional point location is determined by calculating the return time of the reflection from when it was transmitted. Anomalous points can exist, however, and may take the form of points recorded below the ground surface as a result of timing errors in the lidar system due to multiple reflections within the canopy and ground material, or points far above the canopy perhaps due to bird strikes. Often times these points are classified as “Noise” during post-processing, but are not always completely removed. Anomalous points were considered and removed if present by defining valid points as lying between the 1st and 99th percentile of all height values within the point cloud; anomalies were defined as lying outside of those ranges. A comparison showing the point cloud for a single lidar file before and after removing the height anomalies is shown in Fig. 5.

Figure 5 Lidar point cloud showing height anomalies (A) and after height anomalies were removed (B).

The point cloud in (B) can be used to generate a pseudo-waveform feature.

Valid lidar points within each tree crown geometry were used to create a pseudo-waveform for the tree crown which simulates the entire crown’s footprint. Muss, Mladenoff & Townsend, 2011 have shown this representation of the lidar point cloud to give an accurate representation of vegetation structure as a 1-dimensional signal. We define the pseudo-waveform by calculating the density of points within one-meter height bins ranging from zero meters above ground to the maximum height above ground for any given tree in the training data. This resulted in 39 one-meter height bins ranging from 0 to 40 m above ground. Bins with no points were given a point density value of zero. This results in a table where each row represents a 1-d structural signal for each tree crown geometry, which are used as additional features in the fusion network for classification. See Fig. 6 for example pseudo-waveforms and the corresponding point clouds used to generate them.

Figure 6 Two examples of different pseudo-waveforms from individual tree crown geometries.

Original lidar point clouds (A and C) and corresponding pseudo-waveforms (B and D) showing point density at each height bin. Labels are taken from the “indvdID” field from the training data. Note the difference in height values between the two examples.

The fusion network

To learn how to combine information from the RGB, hyperspectral, and lidar data, we concatenated the probability vectors from the RGB CNN step, hyperspectral reflectance spectra at the centroid of each tree crown polygon, and lidar pseudo-waveforms into a feature vector that was passed as input to a neural network (also known as a multilayer perceptron), the so-called “fusion network” (Goodfellow et al., 2016). The fusion network was relatively shallow with two hidden layers (size 64 and 32). The input to the fusion network was a feature vector with 440 elements: 31 class probabilities from the RGB ConvNet (one per taxon code), 369 reflectance values from the hyperspectral data (one per wavelength after “bad bands” with high noise due to water absorption were removed), and 40 features from the lidar data (proportions for 39 bins, and the total number of points across all bins). The output of the fusion network was a concatenated vector of taxon probabilities.

In early versions of our model we noticed a tendency to overpredict the most abundant taxa, a problem which we thought might be related to the default cross-entropy loss. We trained the fusion network by minimizing a custom “soft F1” loss function rather than cross-entropy to try to generate predictions that were robust to class imbalance in the training data. Given a classification task with K classes, a length K vector of probabilities θ and a one-hot-encoded vector y of length K, the soft F1 loss can be computed as: L(θ)=K−1∑k=1K1−2θkyk2θkyk+θk(1−yk)+(1−θk)yk+ϵ,

where ε is a fixed small number (e.g., 1e−7) to prevent division by zero. We used stochastic minibatch gradient descent to minimize the expected soft F1 loss in the training data, using a batch size of 64 examples, averaging loss among examples within each minibatch. The fusion network was trained for 20 epochs using a 1 cycle policy, with a maximum learning rate of 1e−2. The number of epochs and the maximum learning rate were chosen based on our 20% partition of the training data that were set aside as an initial validation set (Smith, 2018).

Post-processing of fusion network output

To deal with out-of-distribution classes (taxa in the test sites that were not in the training data), we decided to place some probability mass on an “other” class when the model predictions were not confident. If the maximum class probability from the fusion network was less than 0.5, we assigned a probability of 0.5 to the “other” class and renormalized the remaining probabilities so that the entire probability vector summed to one. We chose to use a probability threshold of 0.5 based on qualitative visual inspection of the classification probability histograms.

Implementation

We processed the RGB and hyperspectral data using GDAL (GDAL/OGR contributors, 2020) and R (R Core Team, 2020). Specifically, we used the neonhs (Joseph & Wasser, 2020) R package to extract hyperspectral reflectance data at the center of each tree crown polygon. We processed the lidar data in Python (Van Rossum & Drake, 2009). As we split the data generation and processing tasks across members of our team, we worked collaboratively and uploaded files to a shared a Google Drive that was readable from Google Colab (Bisong, 2019). We implemented the CNN and fusion network with fastai in Google Colab (Howard & Gugger, 2020). The code that we developed for our methods is openly available on GitHub (https://github.com/earthlab/idtrees_earthlab) to be freely used and improved upon by the ecological community.

Results

While initially developing and assessing our methods, we withheld 20% the training data as an initial validation set, which contained 206 samples spanning all 31 taxon classes. We created a confusion matrix to assess classification accuracy (Fig. 7).

Figure 7 This confusion matrix compares the true and predicted taxon class labels using our tabular classification model.

The data used here consist of the 20% validation subset from the training data. Counts along the diagonal indicate correct predictions.

Our initial validation set classification accuracy was 0.51. The taxa with the most accurate predictions in descending order were PIPA2, ACRU, QUAL, QURU, QULA2, QUCO2, NYSY, and PIEL, many of which were among the most abundant in the training data set (Table 1).

For the final competition evaluation, we applied our classifier to a test data set without knowing the true taxon labels. We submitted a file with our predicted probabilities that each individual plant in the test set belonged to each taxon class, including an unknown class, “other”. The IDTReeS competition organizers compared our submitted predictions to the true taxon class labels for each tree crown and provided us with a reduced confusion matrix and a corresponding score report based on true class accuracy. The reduced confusion matrix compares the true and predicted labels for each tree, grouping all out-of-sample taxa into a single class called “other” (Fig. 8). This was done to see the direct match between our predictions of the “other” class with the correct label of “other”.

Figure 8 This confusion matrix compares the true and predicted taxon class labels using our tabular classification model.

The data used here consist of the test set for the final competition evaluation. Counts along the diagonal indicate correct predictions.

The score report provided based on our predicted taxon labels includes the following metrics calculated using the scikit-learn Python library (Pedregosa et al., 2011): macro average F1 score, weighted average F1 score, and accuracy score from scikit-learn’s “classification_report”. The Macro Average F1 score considers all predictions from all classes when calculating the F1, whereas the weighted average F1 score considers the relative number of samples per class while computing the F1 score. The accuracy score calculates the global fraction of correct predictions. Our scores for each of these evaluation metrics are summarized in Table 3, with the full set of scores for each species shown in Table 4.

Table 3 Taxon prediction results summarized by competition evaluation metrics.

Evaluation metric	Score	
Macro Average F1	0.07	
Weighted Average F1	0.31	
Accuracy	0.32	
Categorical Cross-Entropy	11.62	

Table 4 Full report of competition classification evaluation metrics. These test set results include the classifier total accuracy, Macro F1 score, and weighted average F1 score in bold.

	Precision	Recall	F1-score	Support	
ACRU	0.15625	0.147059	0.151515	34	
ACSA3	0	0	0	3	
AMLA	0	0	0	0	
CAGL8	0	0	0	19	
CATO6	0	0	0	0	
FAGR	0	0	0	3	
LITU	0	0	0	14	
NYBI	0	0	0	0	
NYSY	0	0	0	12	
OXYDE	0	0	0	0	
Other	0.259259	0.185841	0.216495	113	
PIEL	0	0	0	0	
PINUS	0	0	0	5	
PIPA2	0.65	0.80791	0.720403	177	
PITA	0	0	0	30	
PRSE2	0	0	0	0	
QUAL	0.076923	0.043478	0.055556	23	
QUCO2	0	0	0	0	
QUERC	0	0	0	23	
QUGE2	0.153846	0.1	0.121212	20	
QUHE2	0	0	0	3	
QULA2	0.40625	0.371429	0.38806	35	
QUMO4	0	0	0	15	
QUNI	0	0	0	22	
QURU	0.384615	0.208333	0.27027	24	
ROPS	0	0	0	5	
TSCA	0	0	0	5	
accuracy	0.324786	0.324786	0.324786	0.324786	
macro avg	0.077302	0.069039	0.071241	585	
weighted avg	0.304196	0.324786	0.309226	585	

Discussion

Here we presented our plant taxon classification approach that combines a convolutional neural network (CNN) for RGB images with a downstream fusion network that integrates RGB, hyperspectral, and lidar data. Tree species classification accuracy values vary wildly throughout the literature, based on factors such as the number of species being classified and the types of remote sensing systems that captured the data. For instance, a recent review of 101 studies found reported accuracies ranging from less than 60% to nearly 100% to classify anywhere from a couple to nearly 30 species using combined sensor systems (Fassnacht et al., 2016).

Our classification workflow combined data from all three National Ecological Observatory Network (NEON) airborne remote sensing systems and yielded an overall accuracy of 0.51 for a subset of the training set and 0.32 for the competition test set. The accuracy values that our method achieved are on the low end of the range reported by Fassnacht et al. (2016), although it is worth noting that our method was tasked with classifying 31 species or genera in addition to identifying a series of unknown species in the final competition evaluation, which exceeds the high end of the number of species that the studies classified in the recent review (from a couple to less than 30). For the five participating teams in this IDTReeS competition, the overall classification accuracy values ranged between 0.32 and 0.47, macro average F1 scores ranged between 0.07 and 0.28, weighted average F1 scores ranged between 0.31 and 0.45, and cross entropy scores ranged between 2.5 and 11.62. While our model did well for common classes, poor performance on rare and out-of-distribution classes was a major limitation. The large difference between the macro average F1 score and the weighted average F1 score for the classifier is indicative of the class imbalance and poor classifier performance for rare classes. Table 4 shows the class imbalance present in the test dataset which is reflected in the test dataset confusion matrix, Fig. 8. Based on the confusion matrices from the training set (Fig. 7) and test set (Fig. 8), our model struggled to perform as well at the unseen site and unknown taxon classes. We obtained an overall accuracy of 0.51 when predicting the taxon labels in our 20% withheld from training, which was higher than the overall accuracy of 0.32 reported for the test set, which might be indicative of overfitting on the validation data.

Aside from overfitting, poor performance on out-of-distribution data could be due to dataset shifts or differences at the third NEON site. For instance, we found image artifacts such as distortion or the presence of shadows due to illumination conditions are variable across plots and NEON sites, which are visible in Fig. 3. Lots of distortion was visible in images from the OSBS site, likely an effect of wind during the data collection flight. Note that the presence and appearance of this distortion is not consistent across the images. These artifacts in addition to the highly variable individual tree crown polygons, which we transformed to be squares of uniform size, likely challenged the RGB portion of our classification approach. We discussed the possibility of filtering small or oddly shaped crowns (i.e. one pixel wide by six pixels tall) since these shapes may be due to occlusion by neighboring crowns on a per-case basis, and may not necessarily be representative of that taxon’s typical crown dimensions. However, without doing more in-depth analysis about which shapes or dimensions to filter, we kept all individual tree crown shapes in the data set for our analysis.

Poor performance for out-of-distribution data could also be attributed to uncertainty calibration for the “other” class. Our approach to deal with these unknowns was to use a 0.5 certainty threshold to label an individual as “other”. We correctly identified 21 of 113 trees with true labels of “other”, which amounts to 22% of them. As described in our methods section, our decision to use this 0.5 threshold was based on the distribution of probabilities observed during training. Further tuning or increasing this threshold may lead to better identification of unknown taxon classes in the future.

We spent some time brainstorming different approaches to handle out of distribution classes (taxa present in the test set that were absent in the training data). Our final solution to this (ad hoc “other” class predictions) was a much simpler version compared to some of the ideas that we had. Most elaborate among these abandoned ideas was to use K-fold cross-validation to iteratively generate K train/validation splits of the training data, some of which would result in some taxa being only represented in the validation data. Our thought was to try to build a model that was well-calibrated based on this cross-validation, i.e., a model that was able to predict “other” when presented with a taxon that was not represented in the training data.

Related to predictive features to train the classifier, we investigated texture measures from the RGB data as a potential set of features to use as inputs for classification. Preliminary analysis on Haralick, Shanmugam & Dinstein, 1973), calculated from each tree’s gray-level co-occurrence matrix, did not prove separable at the taxon level when considering the training data. A principal components transform (Rodarmel & Shan, 2002) was applied to the texture feature space, but the transformed axes did not prove separable, either. We also explored dimensionality reduction methods directly with the hyperspectral data, which are commonly used to summarize data from hundreds of highly correlated hyperspectral bands into fewer bands (Fassnacht et al., 2016; Maschler, Atzberger & Immitzer, 2018). Another approach to perform dimensionality reduction would be to use an auto-encoder (Wang, Yao & Zhao, 2016). Including additional descriptive features as a result of dimensionality reduction methods like principal component analysis, spectral indices, or targeted feature selection such as specific spectral bands may improve future classifier efforts. Additional ideas for improving classifier performance in the data preprocessing steps include identifying and removing (or utilizing) non-vegetation and shadow pixels (Mostafa, 2017), which are especially visible in the high spatial resolution RGB images (Fig. 3).

We made use of the lidar point cloud data by resampling the points into pseudo-waveforms, which allowed us to incorporate information about point density at different heights within the canopy. Future classification methods may benefit from incorporating additional point cloud-derived metrics, such as modeling the shape of the crown, distances between first and last returns, as well as intensity information, although this may require data with a higher point density (Korpela et al., 2010). The only competition dataset that we did not incorporate into our classifier was the rasterized lidar-derived canopy height model (CHM) (Goulden & Scholl, 2019). The CHM data was at 1 meter per pixel resolution and we felt that it did not provide enough information relative to the other datasets, particularly its cohort of the feature-rich 1 meter hyperspectral imagery. Thus, we made an executive decision to not include the data as the boundaries of the tree crowns as observed in the RGB data (Fig. 3) were much too coarse to justify using the CHM as a means by which to crop any of the other data. However, with higher resolution CHM data or larger crown geometries, we foresee being able to directly use the CHM information about crown geometry to generate better data subsets and extractions for individual trees (Scholl et al., 2020).

Early on in the competition, we discussed the merits of a one- vs two-stage approach for data integration. While we settled on a two-stage approach (CNN to fusion network), a one-stage approach might have been a viable option. In a one-stage approach, we would embed the CNN within the fusion network, and instead of passing the output to a downstream model, we would concatenate the feature vector generated from the convnet with the vector valued features in the fusion network to obtain a model that is end-to-end differentiable. It was not clear that this would result in a better model, but it was clear it would require considerably more effort.

Conclusions

The IDTreeS 2020 plant classification challenge openly invited teams to create and compare their methods using open-source NEON data. In this paper, we presented the methods and results of the team called Jeepers Treepers. We used a two-stage deep learning fusion network approach to combine features from RGB, hyperspectral, and lidar point cloud data to classify taxa at an unseen site featuring unknown species. Creating classification methods that are transferable and generalizeable is no easy task, which made it an interesting topic for this data competition. Overall, we believe that further processing and filtering the RGB images (such as calculating texture metrics and manually removing images containing notable image artifacts or non-vegetation pixels), refining the logic for identifying unknown taxa (when assigning individuals to the “other” class), further addressing the taxon imbalance in the training data set, and incorporating greater data volume and features (such as additional lidar point cloud metrics based on point height and intensity) would improve our classifier’s performance. We see value in the open data-driven competition format to accelerate methods development in the computational ecology field, and encourage others to participate in the future.

We would like to thank the members of the Earth Lab Deep Learning meet up group for brainstorming about our methods during Spring 2020. We also appreciate the IDTReeS competition organizers’ flexibility and responsiveness during the unprecedented year of 2020.

Additional Information and Declarations

Competing Interests

Author Contributions

Data Availability

The authors declare that they have no competing interests.

Victoria M. Scholl conceived and designed the experiments, performed the experiments, analyzed the data, prepared figures and/or tables, authored or reviewed drafts of the paper, and approved the final draft.

Joseph McGlinchy conceived and designed the experiments, performed the experiments, analyzed the data, prepared figures and/or tables, authored or reviewed drafts of the paper, and approved the final draft.

Teo Price-Broncucia conceived and designed the experiments, performed the experiments, analyzed the data, authored or reviewed drafts of the paper, and approved the final draft.

Jennifer K. Balch analyzed the data, authored or reviewed drafts of the paper, and approved the final draft.

Maxwell B. Joseph conceived and designed the experiments, performed the experiments, analyzed the data, prepared figures and/or tables, authored or reviewed drafts of the paper, and approved the final draft.

The following information was supplied regarding data availability:

The data is available on Zenodo: Graves, Sarah, & Marconi, Sergio. (2020). IDTReeS 2020 Competition Data (Version 4) [Data set]. Zenodo. DOI 10.5281/zenodo.3700196.

The code is available at GitHub: https://github.com/earthlab/idtrees_earthlab.

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
