# Peer review of "Fusion neural networks for plant classification: learning to combine RGB, hyperspectral, and lidar data"

_PeerJ, doi:10.7717/peerj.11790_

## Round 0.1 · original submission · Major Revisions

As you can see from the Reviewers’ comments below, the opinions of the Reviewers range from very positive to critical. The criticism of Reviewers 1 and 2 is mainly towards the low classification and out-of-sample accuracy obtained in the study. However, it is clear that the manuscript acknowledges and addresses this issue, and provides valuable information for future method development in the computational ecology field. Based on this and the PeerJ Editorial Criteria I have decided that the manuscript can be further considered for publication in PeerJ. Please address any of the Reviewers' concerns that have not already been explained and discussed in the manuscript and provide a point-by-point response to the Reviewers' comments in your rebuttal letter.

Reviewer 1 ·

Basic reporting

no comment

Experimental design

The experimental design lacks a lot of data preprocessing, resulting in a low classification accuracy of 0.51.

Validity of the findings

no comment

Additional comments

This study aims to combine RGB, hyperspectral, and lidar data for plant classification. They developed a two-stage deep learning approach to integrate NEON remote sensing data from all three sensors and classify individual plant species and genera. The first stage was a convolutional neural network that generates taxon probabilities from RGB images, and the second stage was a fusion neural network that "learns" how to combine these probabilities with hyperspectral and lidar data. This thought is promising. However, they only obtained an overall classification accuracy of 0.51. The authors even pointed out in the discussion that their accuracy is about the lowest in 101 studies reported in a recent review. Such low accuracy cannot make readers willing to use the approach proposed in this study. It seems that the authors didn’t find the best input for the neural network. Besides, they didn’t remove the image artifacts. Therefore, I cannot recommend publishing the manuscript in its current form.

Reviewer 2 ·

Basic reporting

no comment

Experimental design

no comment

Validity of the findings

no comment

Additional comments

This article introduced a two-stage approach for tree species classification which combined high-res imagery hyperspectral and lidar point cloud: first using a pre-trained convolutional neural network to generate probability images from high-res imagery, and then applied a fusion neural network with hyperspectral and lidar data. Though the study is complete and is somewhat innovative on combining different sources, the out-of-sample accuracy reported in the article is too low (0.32), and hence the proposed approach is not considered to be potentially applicable. Even after excluding the hardest class ‘other’, categorical f1-score were zero still for many species. I encouraged the authors to resubmit your article after reaching an improved out-of-sample accuracy (at least 0.5 for out-of-sample overall accuracy).

As mentioned in the discussion part, there are several potential improvements for increasing the accuracy. The first idea bumping to my brain, as you also mentioned, is to increase the certainty threshold, considering your model didn’t perform well for the ‘other’ class. Besides, I suggested taking better use of hyperspectral images. An input of comprehensive hyperspectral bands is not acute enough for capturing subtle spectral differences among species. Feature selection is warranted before fusion neural network model.

The remote sensing community has been struggling with the categories which are not defined in the classification scheme (i.e., ‘other’) for decades. To me, this problem might be already out of technical scope. You may need to re-define the classification scheme to narrow the concept of ‘other’ categories, considering the category for ~20% in your out-of-sample set is ‘unknown’. If the competition organization allows, I would recommend you to re-select your training sample set which can be more representative of your complete classification scheme.

Reviewer 3 ·

Basic reporting

The paper was quite clearly written.

Intro and background showed context etc…

Good structure, raw data is shared (I think)

Figures were relevant, high quality and well captioned

I think they tell where to find data and code.

Experimental design

Original, and definitely within scope of journal.

The research question is very clear, as it is a competition.

It is rigorous, and they describe where assumptions are made and potential shortfalls in rigour exist.

Methods are described in detailed easily understood language, better than most papers I have read honestly.

Validity of the findings

Impact and novelty not accessed ( i love this criteria, it is annoying when authors talk about why their work is great). They do a good job assessing where the model could be improved.

Underlying data comes from he IDTrees competition, so yeah, I guess its sound.

Speculation is mostly about shortcoming in the model and is clearly identified as such.

Conclusions are well stated etc..

Additional comments

Overall, very interesting and well written paper. I recommend for publication with minor revisions.

Sentence beginning on line 162: The laser penetrates the canopy …
On my first read I found this wording to be a bit hard to follow, though now as i look at it maybe its not so bad.

Sentence beginning on line 180: If the Maximum…
This sentence could be more clear. I think that they are saying that the bins above the tree are filled with zeros, which is what I would expect, but the phrasing caused me confusion.

Sentence beginning on line 212: The decision to use…
This sentence struck me as kind of funny, it seems kind of like, we just picked a number while gazing at an image. It's probably fine, but that is how it came across to me.

---

## Round 0.2 · accepted · Accept

Thank you for providing thorough responses to the Reviewers' comments and revising the manuscript accordingly. The intent of the article, i.e., to report on the success accomplished by competing research teams in a short period of time to better refine methodological approaches for species classification, has been clearly explained and the manuscript can be accepted for publication.